# Comparison of In-Frame Deletion, Homology-Directed Repair, and Prime Editing-Based Correction of Duchenne Muscular Dystrophy Mutations

**DOI:** 10.3390/biom13050870

**Published:** 2023-05-22

**Authors:** Xiaoying Zhao, Kunli Qu, Benedetta Curci, Huanming Yang, Lars Bolund, Lin Lin, Yonglun Luo

**Affiliations:** 1College of Life Sciences, University of Chinese Academy of Sciences, Beijing 101408, China; zhaoxiaoying@genomics.cn; 2Lars Bolund Institute of Regenerative Medicine, Qingdao-Europe Advanced Institute for Life Sciences, BGI-Qingdao, BGI-Shenzhen, Qingdao 266555, China; qukunli@genomics.cn (K.Q.); yanghm@genomics.cn (H.Y.); bolund@biomed.au.dk (L.B.); 3Department of Biomedicine, Aarhus University, 8000 Aarhus, Denmark; benedetta.curci@outlook.it (B.C.); lin.lin@biomed.au.dk (L.L.); 4Department of Biology, Copenhagen University, 2200 Copenhagen, Denmark; 5HIM-BGI Center, Hangzhou Institute of Medicine (HIM), Chinese Academy of Sciences, Hangzhou 310022, China; 6Steno Diabetes Center Aarhus, Aarhus University Hospital, 8200 Aarhus, Denmark

**Keywords:** gene editing, CRISPR, Cas9, prime editing, DMD, gene therapy, reporter vector

## Abstract

Recent progress in CRISPR gene editing tools has substantially increased the opportunities for curing devastating genetic diseases. Here we compare in-frame deletion by CRISPR-based non-homologous blunt end joining (NHBEJ), homology-directed repair (HDR), and prime editing (PE, PE2, and PE3)-based correction of two Duchenne Muscular Dystrophy (DMD) loss-of-function mutations (c.5533G>T and c.7893delC). To enable accurate and rapid evaluation of editing efficiency, we generated a genomically integrated synthetic reporter system (VENUS) carrying the *DMD* mutations. The VENUS contains a modified enhanced green fluorescence protein (*EGFP*) gene, in which expression was restored upon the CRISPR-mediated correction of *DMD* loss-of-function mutations. We observed that the highest editing efficiency was achieved by NHBEJ (74–77%), followed by HDR (21–24%) and PE2 (1.5%) in HEK293T VENUS reporter cells. A similar HDR (23%) and PE2 (1.1%) correction efficiency is achieved in fibroblast VENUS cells. With PE3 (PE2 plus nicking gRNA), the c.7893delC correction efficiency was increased 3-fold. Furthermore, an approximately 31% correction efficiency of the endogenous DMD: c.7893delC is achieved in the FACS-enriched HDR-edited VENUS EGFP+ patient fibroblasts. We demonstrated that a highly efficient correction of *DMD* loss-of-function mutations in patient cells can be achieved by several means of CRISPR gene editing.

## 1. Introduction

Since the first report in 2012, the Clustered Regularly Interspaced Short Palindromic Repeats (CRISPR) and CRISPR-associate protein 9 (Cas9) gene editing tools have been largely expanded [1]. Conventional CRISPR/Cas9-based gene editing relies on the introduction of a double-strand break (DSB) in the genomic target site, which is repaired by the intrinsic DSB repair mechanisms in cells. In mammalian cells, the CRISPR-induced DSB is predominantly repaired by the error-prone non-homologous end joining (NHEJ) and microhomology-mediated end joining (MMEJ) pathways, leading to the introduction of small deletions or insertions, collectively known as indels, at the DSB site [2]. However, we previously observed that when two DSBs are simultaneously generated by CRISPR on the same chromosome, these two DSBs are predominantly repaired by blunt end joining without introducing other indels [3]. We defined this type of repair event as non-homologous blunt end joining (NHBEJ), providing an efficient and alternative approach to predictive deletion editing and CRISPR therapy.

Another DNA repair mechanism is homology-directed repair (HDR). Compared to NHEJ and MMEJ, HDR efficiency is much lower [4]. In our experience, it is 30 to 100 folds lower, depending on the editing sites and HDR DNA templates. However, there still is a substantial need for gene editing technologies for studying or correcting pathogenetic variants. Millions of clinical variants with known or unclear pathogenic functions have been reported in the ClinVar database (https://www.ncbi.nlm.nih.gov/clinvar/ (accessed on 21 February 2023)). Over the last decade, two types of CRISPR-derived technologies have been developed for site-specific editing that are independent of DSB and DNA repair templates. By fusing an engineered deaminase to the Cas9 in which DNA nuclease domains are partially (Cas9 nickase, nCas9) or completely (dead Cas9, dCas9) inactivated, a serial of so-called base editing technologies (e.g., adenine base editors (ABE), cytosine base editors (CBE), and glycosylase base editors (GBE)) has been developed for inducing DNA substitution (transitions or transversions) [5,6,7]. The other type of CRISPR-derived gene editing technology is prime editing (PE), which is a precise gene editing technique that can mediate targeted deletion, insertion, and base substitution without introducing DSBs or DNA templates. Prime editing is developed by fusing a reverse transcriptase (RT) to the nCas9(H840A) [8]. Unlike the guide RNA (gRNA) used by conventional Cas9 or Cas9-derived base editors, the prime editing gRNA (pegRNA) contains extended RNA sequences in its 3′ ends. A primer binding site (PBS) at the 3′ end of the pegRNA binds to the 3′ flap of the non-target strand generated by the nCas9(H840A) and initiates the reverse transcription process. Immediately upstream of the PBS, a reverse transcription (RT) RNA template containing the desired edit is transcribed, thus generating a 3′ flap containing the desired edit that is permanently inserted into the target site through flap resolution. The efficiency of prime editing 2 (PE2), which uses nCas9(H840A)-RT and pegRNA, can be further increased by introducing a proximal nick in the other strand with a second gRNA (PE3). Prime editing is a promising gene editing approach for recoding and correcting the indel mutations in cells.

Duchenne Muscular Dystrophy (DMD) is an X-linked recessive and muscle-wasting disorder, which is the most prevalent childhood muscular dystrophy and affects approximately 1 in 5000 male newborns worldwide [9]. DMD is caused by loss-of-function mutations in the *DMD* gene (gene length = ~2.3 Mb, 79 exons), which encodes the dystrophin protein. Dystrophin is part of the dystrophin–glycoprotein complex and acts as a link between the actin cytoskeleton and the extracellular matrix in muscle cells, providing membrane stability. To date, many disease-causing mutations (including exon deletion, exon duplication, small deletions/insertions, missense mutation, splicing mutation, and nonsense mutation) have been detected in the *DMD* gene. Certain regions of the *DMD* gene were found to be mutational “hotspots”. Around 60% of DMD cases were caused by mutations located in/around exons 45 to 55. Exons 2–10 are also among the highly mutated regions [10]. All these loss-of-function (LOF) *DMD* mutations result in a version of dystrophin that does not function correctly or the loss of dystrophin expression entirely, ultimately causing DMD. Currently, there is no cure for DMD. However, to restore muscular function in DMD patients, numerous treatment approaches have been explored, including using CRISPR editing to correct the *DMD* mutation and restore dystrophin expression [11].

Several methods have been developed for analyzing CRISPR-induced indels in cells. For a complete overview of these methods, we refer readers to a recent review by Eric and colleagues [12]. Previously, we developed a dual fluorescence reporter vector, C-Check, for the quantification of CRISPR gRNA cleavage efficiency based on single-strand annealing, which mediated the repair of the CRISPR-induced DSB [13]. However, this reporter system is not suitable for quantifying the restoration efficiency of loss-of-function (LOF) mutation. In this study, we assessed the efficiency of correcting DMD LOF mutations in cells by three CRISPR gene editing strategies using a genomically integrated engineered EGFP vector (VENUS). We engineered two VENUS vectors carrying a DMD: c.5533G>T (nonsense mutation) and a DMD: c.7893delC (frameshift mutation), respectively. The efficiencies of DMD mutation correction by three CRISPR gene editing approaches, including NHBEJ, HDR, and PE (PE2 and PE3), were quantified by FACS and sequencing in HEK293T and DMD patient fibroblasts. We demonstrated that highly efficient correction of LOF DMD mutations can be achieved with CRISPR gene editing tools.

## 2. Materials and Methods

### 2.1. Oligos and Synthetic gRNAs

PCR primers and VENUS oligonucleotides are ordered from Merck KGaA, Darmstadt, Germany. The ssODN (single-stranded oligo deoxynucleotides) for HDR and pegRNA are ordered from Integrated DNA Technologies (IDT, Coralville, IA, USA). Chemically modified synthetic CRISPR gRNA is ordered from Synthego, Redwood City, CA, USA.

### 2.2. Design of VENUS Oligos

For the VENUS-DMD:c.5533G>T vector, 42 base pairs of the DMD coding sequencing (reading frame from +1), including the DMD:c.5533G>T, were selected. For the VENUS-DMD:c.7893delC vector, 41 base pairs of the DMD coding sequencing (reading frame from +1), including the DMD:c.7893delC, were selected. A “5′-CTAGgg” and a “5′-CTAG” overhang sequence was added to the sense and antisense VENUS oligonucleotides, respectively. A conventional VENUS oligo design sheet is provided in Appendix A.

### 2.3. Construction of VENUS Vectors

A dicistronic lentiviral vector pCCL-PGK-EGFP-IRES-Puro (a gift kindly provided by Prof. Jacob Giehm Mikkelsen from Aarhus University), which contains an XbaI cloning site between the ATG start codon and another coding sequencing of the enhanced green fluorescence protein (*EGFP*) gene. To generate the VENUS reporter vector, pCCL-PGK-EGFP-IRES-Puro was digested with XbaI (Thermo Fisher Scientific, Waltham, MA, USA, FD0684) followed by Calf Intestinal (CIP) alkaline phosphatase treatment to remove 5′-phosphate groups from the linearized plasmid. The linearized plasmid was purified with 1% agarose gel.

To generate the double-stranded VENUS DNA, 1 µL of each sense and anti-sense VENUS oligonucleotide (100 µM) was mixed in a 2 µL 10× NEB buffer 2 and 16 µL ddH_2_O. The two oligonucleotides were first annealed by denaturing at 95 °C for 5 min, followed by slowly cooling down to 25 °C at a rate of −5 °C/min.

For ligation of the double-stranded VENUS DNA into the linearized pCCL-PGK-EGFP-IRES-Puro vector, a 200 ng backbone and 2 µL of annealed VENUS DNA were ligated with 1 µL of T4 ligase (Thermo Fisher Scientific, EL001) in a 1 µL T4 ligase buffer in a 10 µL reaction. Ligation was performed at 16 °C for 4 h. After ligation, a 10 µL ligation product was transformed into 50 µL competent *E. coli* cells and grown in an LB agar plate with Ampicillin overnight.

We adopted a PCR screening strategy to screen for *E. coli* colonies carrying the correct VENUS plasmid. We used the VENUS sense oligonucleotide as a forward primer and an internal primer located in the antisense strand of the *EGFP* coding region as a reverse primer (Appendix A). Each PCR reaction (15 µL) contains a 1 µL *E. coli* lysate, 0.08 µL Dream Taq (Thermo Fisher Scientific, EP0702), 0.6 µL forward (5 µM), 0.6 µL reverse primer (5 µM), 0.3 µL dNTP (10 mM), 1.5 µL 10× Dream Taq buffer, and ddH_2_O. PCR was performed with a program of one cycle at 94 °C for 2 min, 35 cycles at 94 °C for 20 s, 62 °C for 30 s, and 72 °C for 40 s, and 1 cycle at 72 °C for 7 min. PCR-positive colonies were further grown in an LB medium and the VENUS plasmid was purified by a PureLink Hipure Plasmid Maxiprep kit (Thermo Fisher Scientific, K210003), followed by Sanger sequencing to validate the correct insert.

### 2.4. Lentivirus Packaging

HEK293T cells were cultured in a DMEM Medium with 10% FBS (D10). One day before transfection, 2 × 10^6^ HEK293T cells were seeded in a P10 Petri dish, which reached approximately 70% confluence on the day of transfection. For each lentivirus packaging transfection, each transfection mix contains 66 µL PEI (1 µg/µL), 8.6 µg VENUS plasmid, 2.5 µg pMD.2G, 8.6 µg pRRE, and 2 µg pRSV-REV in 1 mL OptiMEM. The transfection cocktail was mixed well by gently pipetting, incubating at room temperature for 15 min, and then adding it into a 10 cm dish dropwise. After 48 hr of transfection, the cell culture medium was collected, and cell debris was removed with a 0.45 µM filter. Polybrene was added to the crude lentivirus supernatant to the final concentration of 8 µg/mL, which was aliquoted into 1.5 mL EP tubes (1 mL per tube) and stored at −80 °C before use.

### 2.5. Generation of Genomically Integrated VENUS Cell Line

A wild type HEK293T cell and DMD fibroblasts were seeded into a 6-well plate (3 × 10^5^). At a cell confluency of 50–70% (approximately 24 h after cell seeding), cells were changed to a fresh DMEM medium with 10% FBS (D10) and 8 μg/mL of polybrene, and 200 µL of a crude virus was added to each well. After 48 h of transduction, cells were changed to a D10 medium with 1 µg/mL of puromycin (D10-Puro1), hereafter refreshed with a D10-Puro1 medium every two days. Cell death was observed three days after culturing cells in the D10-Puro1 medium. When all cells in the un-transduced control group were dead, Puro-resistant VENUS-transduced cells were cultured in D10-Puro0.5 (0.5 µg/mL puromycin) for maintenance and expansion.

### 2.6. Cell Culture

Wild type HEK 293T cells were cultured in DMEM with 10% FBS and 1% P/S. HEK 293T cells with VENUS were grown in DMEM with 10% FBS, 1% P/S, and 0.5 µg/mL puromycin. DMD Fibroblast was grown in DMEM F-12 with 20% FBS, 1% P/S, and bFGF (5 ng/mL). Fibroblast with VENUS was grown in DMEM F-12 with 20% FBS, 1% P/S, bFGF (5 ng/mL), and 0.5 µg/mL puromycin. All cells were passaged when confluency reached 80%.

### 2.7. Nucleofection

We used a Lonza 4D nucleofection for all CRISPR nucleofection. All gRNAs were dissolved in nuclease-free water to a concentration of 100 pmol/µL.

NHBEJ: 0.6 µL Cas9 protein (IDT), 0.6 µL Sp1 gRNA, 0.6 µL Sp2 gRNA, and 16.2 µL Opti-MEM were mixed in an EP tube and incubated at room temperature for 15 min. Then, 200,000 cells were resuspended with the 18 µL ribonucleoprotein (RNP) mixture and transferred to a 16-well nucleofection chip (Lonza). Nucleofection was performed with the X channel CM138 program.

HDR: RNP was prepared as NHBEJ, except for 0.3 µL ssODN (100 pmol/µL), which was added to the RNP mixture after 10 min of incubation at R.T.

PE: PE2 mRNA was generated from the pCMV-PE2 plasmid using an in vitro transcription kit (see below). The pCMV-PE2 was a gift from David Liu (Addgene plasmid # 132775; http://n2t.net/addgene:132775/ (accessed on 15 March 2022); RRID:Addgene_132775). For each nucleofection, 0.8 µL pegRNA (100 pmol/µL) and 2 µL PE2 mRNA (1 µg/µL) were mixed with Opti-MEM to a total volume of 18 µL. PE3 uses the same 0.8 µL pegRNA (100 pmol/µL) and 2 µL PE2 mRNA (1 µg/µL) with PE2 and was then mixed with another 0.8 µL nicked gRNA (100 pmol/µL) in Opti-MEM to a total volume of 18 µL.

### 2.8. In Vitro Transcription (IVT)

A 10 µg pCMV-PE2 plasmid was digested with 5 µL PmeI (FastDigest, FD1344) in a total volume of 50 µL and purified with a PCR clean-up kit (NucleoSpin, #740609.250). IVT was performed using the MEGAscript™ T7 Transcription Kit (AM1334). Each reaction contains an 11 µL dNTP mixture (2 µL ATP (75 mM), 2 µL CTP (75 mM), 2 µL UTP (75 mM), 2 µL GTP (15 mM), and 3 µL Clean Cap), 2 µL 10× reaction buffer, 1 µg linearized DNA template, 2 µL enzyme mix, and NF-H_2_O at a total volume of 20 µL. The IVT reaction was performed by incubating the mixture at 37 °C for 4 h.

Following the IVT reaction, polyA was added to the mRNA using a Poly(A) Tailing kit. The reaction mixture containing the 10 µL 5X E-PAP buffer, 5 µL ATP (10 mM), 8 µL NF-H_2_O, 5 µL MnCl_2_ (25 mM), and 2 µL E-PAP enzyme was added to IVT product, mixed well, and incubated at 37 °C for 1 h. One µL TURBO DNase was then added to the reaction and incubated for 15 min to remove the DNA template.

The IVT mRNA was purified by lithium chloride precipitation. A total of 30 μL of nuclease-free water and 30 μL of a LiCl precipitation solution were added to the IVT reaction. The reaction was mixed well by pipetting gently and placed at −20 °C for at least 30 min. After that, the reaction mixture was centrifuged at 4 °C for 15 min at maximum speed. The supernatant was carefully removed without disturbing the RNA pellet. The RNA pellet was washed once with 1 mL 70% ethanol, resuspended in 30 μL of nuclease-free (NF) H_2_O, and stored at −80 °C.

### 2.9. Fluorescence Imaging

Fluorescence imaging was performed using the ZOE Fluorescent Cell Imager, and three fields of view were randomly selected for each group.

### 2.10. Flow Cytometry (FCM) Analysis

After 72 h of nucleofection, cells were collected for flow cytometry (Quanteon analyzer) to analyze the percentage of GFP positive cells. Data analysis was performed using the NovoCyte analyzer and Quanteon analyzer, and fluorescence channels were selected for GFP wavelengths. Each sample was assayed at 100 µL and the flow analysis results were analyzed using NovoExpress. At least 10,000 cells were analyzed for each experimental replicate.

### 2.11. FACS Cell Sorting

Cell sorting was performed using a 4 Laser FACS Aria III. The fluorescence channel was selected to the fluorescence wavelength of GFP, and all the cells expressing green fluorescence were sorted out and inoculated in 48-well plates for further expansion.

### 2.12. PCR Analysis

Genomic DNA was purified from cells using the NucleoSpin DNA RapidLyse kit (#740100.50). A total of 100 ng of genomic DNA was used as a template for PCR reactions using Dream Taq (Thermo Fisher Scientific, EP0702). Each PCR reaction contains 0.08 µL Dream Taq, with 0.6 µL forward and 0.6 µL reverse primer (5 µM), 0.3 µL dNTP (10 mM), 1.5 µL 10× Dream Taq buffer, 3 µL Betaine (5 M), and 15 µL reaction volume. The PCR program used for VENUS genotyping is 94 °C for 2 min, 35 cycles at 94 °C for 20 s, 62 °C for 30 s, and 72 °C for 40 s, and 1 cycle at 72 °C for 7 min. Genotyping of the endogenous DMD locus in the fibroblasts was performed with pfx polymerase (each reaction contains 0.4 µL pfx enzyme, 100 ng DNA template, 3 µL forward primer and 3 µL reverse primer, and 5 µL 10× pfx reaction buffer with ddH_2_O to total volume as 50 µL) and a PCR program (1 cycle at 95 °C for 2 min, 35 cycles at 95 °C for 15 s, 54 °C for 30 s, and 68 °C for 1 min, and 1 cycle at 68 for °C for 5 min).

### 2.13. Sanger Sequencing

All Sanger sequencing was carried out using the Mix2Seq kit (Eurofins genomics, Ebersberg, Germany).

### 2.14. Statistics

All experiments in this project were performed in three replicates. The data from the FCM in this project were analyzed using NovoExpress software. Prism 7 was used to analyze the data to make graphs and dot plots. The editing efficiency of different editing methods in the VENUS system was calculated using the percentage of cells with green fluorescence detected by flow analysis. One-way ANOVA and Student’s *t*-test were used to compare differences between groups. SnapGene Viewer was used to visualize Sanger sequencing results, and the web tools ICE Analysis (https://ice.synthego.com/#/, accessed on 1 September 2022) and DECODR [14] Analysis (https://decodr.org/, accessed on 1 September 2022) were used for genotype percentage analysis of Sanger data.

## 3. Results and Discussion

To enable the accurate and rapid quantification of LOF correction efficiency, we first developed a complementary reporter system, which can be used for the rapid and sensitive quantification of recoding efficiency to restore loss-of-function (LOF) mutations (nonsense mutation and frameshift indel mutations) in cells. To this end, we generated an engineered-enhanced green fluorescent protein (EGFP) reporter vector encompassing a user-desired LOF mutation in its 5′ coding region (Figure 1A), hereafter called VENUS. The VENUS reporter gene is dicistronic, coding an engineered LOF-EGFP and a puromycin resistance marker through an internal ribosomal entry site sequence (IRES). We generated two VENUS vectors (VENUS-DMDc.5533G>T and VENUS-DMD.c7893delC) carrying the LOF and pathogenic DMD:c.5533G>T (Figure 1B) and DMD:c.7893delC (Figure 1C) mutations, respectively. This was achieved by cloning a small fragment of the DMD coding region (reading frame from +1) containing the LOF mutation into the VENUS vector. As this is an essential step for generating the VENUS system, a separate VENUS design sheet is provided in Appendix A.

We next evaluated three CRISPR-based methods for the correction of DMD LOF mutations using the VENUS system (Figure 2A). NHBEJ is based on the predictive deletion of DNA encompassing the LOF mutations to restore gene expression [3]. It has been shown that reading frame restoring by exon skipping or deletion can restore the majority of the dystrophin protein functions in cells [11,15]. According to the TREAT-NMD DMD Global database, most reported (80%) DMD mutations are large mutations (deletion or duplication of one exon or more) [16], making NHBEJ an attractive approach for DMD therapy. The other two CRISPR editing methods are HDR and PE. These two methods are highly relevant for the correction of small indel mutations and point mutations (Figure 2B).

Instead of using episomal plasmids, we first stably integrated the VENUS reporter vector into the genome of HEK293T cells by lentiviral transduction and puromycin selection (Figure 3A). This ensures that the reporter VENUS locus is within the chromosome/chromatin territory and better mimics the DNA repair processes, as well as the effect of chromatin structure on CRISPR gene editing efficiency, as demonstrated by us previously [17]. We then evaluated NHBEJ, HDR, and PE2-mediated correction of the DMD:c.5533G>T mutation in the HEK239T VENUS-DMD:c.5533G>T reporter cells. Convincingly, EGFP-positive (EGFP+) cells can already be detected 24 h after NHBEJ and HDR editing (Figure 3B). Very few EGFP+ cells were observed in the PE2-edited cells. We quantified the EGFP+ cells 72 h after CRISPR editing with FACS (Appendix A). Our results showed that 74.6 ± 0.3% (n = 3) of NHBEJ (Sp1 + Sp2)-edited cells are EGFP+, which is significantly higher (*p* < 0.0001, one-way ANOVA) than HDR (Sp3 + ssODN1; efficiency = 21.1 ± 0.7%) and PE2 (pegRNA1, efficiency = 0.02 ± 0.01%).

We next conducted similar experiments to restore the reading frame caused by the DMD.c7893delC mutation in the HEK239T VENUS-DMD.c7893delC reporter cells. 77.4 ± 0.3% and 25.0 ± 0.5% of the HEK239T VENUS-DMD.c7893delC were EGFP+ 72 h after NHBEJ (Sp1 + Sp2) and HDR (Sp4 + ssODN2) editing, respectively (Figure 3C and Appendix A). This is similar to the NHBEJ and HDR efficiencies achieved in the HEK239T VENUS-DMD:c.5533G>T cells. However, the PE2 editing efficiency for DMD.c7893delC (pegRNA2) is seven folds higher than DMD:c.5533G>T (pegRNA1) (Figure 3C). This observation is in line with the already reported feature of PE, which exhibits highly variable editing rates between sites and genetic backgrounds [8]. PE2 efficiency can be increased by including an extra nicking gRNA (PE3, nicking gRNA at the endogenous site). Our VENUS system consistently showed that PE3 can increase by three folds (efficiency PE2 = 1.84 ± 0.22, PE3 = 6.99 ± 1.47; *p* = 0.0002 (ANOVA)) for the DMD.c7893delC (pegRNA2) (Figure 3B,C).

Since EGFP expression only reflects the correction of the nonsense mutation for DMD:c.5533G>T and reading frame restoring for DMD:c.7893delC, we next investigated the CRISPR-mediated editing at the VENUS site by target PCR amplification and Sanger sequencing. For NHBEJ-edited cells, we analyzed all cells without a FACS enrichment of EGFP+ cells. Upon successful NHBEJ editing with the Sp1 and Sp2 gRNAs, 87bp and 86bp are deleted from the coding region of the VENUS-DMD:c.5533G>T and VENUS-DMD:c.7893delC reporter genes, respectively (Figure 4 and Figure 5). Sanger sequencing of the NHBEJ PCR product showed the expected blunt end joining junction without additional indels (Figure 4A and Figure 5A). For HDR and PE2-edited HEK293T VENUS reporter cells, EGFP+ cells were first enriched by FACS, followed by a Sanger sequencing analysis of the editing outcomes. In the HEK293T VENUS-DMD:c.5533G>T cells, HDR with Sp3 gRNA and ssODN1 almost completely corrected the DMD:c.5533G>T mutation in the VENUS locus of the EGFP+ cells (Figure 4B). We did not obtain enough EGFP+ cells from the PE2-edited VENUS-DMD:c.5533G>T for a Sanger sequencing analysis due to low editing efficiency (Figure 4B).

In the EGFP+ HEK293T VENUS-DMD:c.7893delC cells, all VENUS loci are edited after HDR with Sp4 gRNA and ssODN2. However, we detected two alleles from Sanger sequencing (Figure 5B). The major allele is repaired by HDR with ssODN2, which recodes the VENUS-DMD:c.7893delC by inserting a C. Another minor allele is the deletion of two base pairs (CA), which also restored the VENUS reading frame. PE2 did not correct all VENUS-DMD:c.7893delC sites in the EGFP+ cells (Figure 5B). A fraction of unedited DMD:c.7893delC was detected by Sanger sequencing in the VENUS loci. Since the HEK293T cells might carry more than one copy of the VENUS-DMD:c.7893delC gene in one cell, it is not surprising to detect both the recoded and unedited alleles in the EGFP+ cells.

Collectively, our results show that VENUS is a sensitive method for analyzing NHBEJ, HDR, and PE2-mediated correction of LOF mutations in cells. Our results also highlight the different editing rates between CRISPR editing methods.

In addition to its utilities as tools for evaluating CRISPR editing efficiency, fluorescence reporter vectors with surrogate editing sites have been used to enrich edited cells [13,18,19,20,21]. We next sought to investigate if the VENUS can be used in other cell types and for the enrichment of gene-edited cells. Since pegRNA1 exhibited very low PE efficiency in the DMD:c.5533G>T, we focused on the DMD:c.7893delC mutation and inserted the VENUS-DMD:c.7893delC reporter vector into the genome of a DMD patient fibroblast (GM05263) carrying the corresponding DMD:c.7893delC in the endogenous DMD gene. Since the NHBEJ Sp1 and Sp2 gRNAs can only target the VENUS locus in the endogenous DMD, we only evaluated the HDR and PE2-mediated correction of the DMD:c.7893delC mutation in the VENUS and corresponding endogenous locus in the GM05263 VENUS-DMD:c.7893delC fibroblasts.

EGFP+ fibroblasts were detected by fluorescence microscopy and FACS three days after the HDR and PE2 editing (Figure 6A,B). Quantification of EGFP+ fibroblasts with flow cytometry showed that 23.3 ± 2.1% and 1.1 ± 0.05% of the HDR and PE2-edited cells were EGFP+ (Figure 6C), an editing rate resembling what was achieved in HEK293T cells (Figure 3C). We FACS-enriched EGFP+ fibroblasts (Figure 6D) and analyzed the genotype at the VENUS-DMD:c.7893delC and endogenous DMD:c.7893delC loci using Sanger sequencing (Figure 6E). Similar to the HEK293T VENUS reporter cells, the VENUS reporter locus in the EGFP+ fibroblasts comprise two major alleles (recode and delCA) after HDR editing. Whereas, in the PE2-edited EGFP+ cells, only part of VENUS-DMD:c.7893delC alleles are recoded. In the EGFP+ HDR-edited cells, 31% of the endogenous DMD.c7893delC were correctly recoded and the second major indel was delCA (10%), which exactly resembled the two major indels recapitulated at the VENUS reporter locus (Figure 6E,F). This agrees with reports using genomically integrated surrogate sites to capture on-target and off-target CRISPR editing efficiency, as well as indel outcomes [22,23,24,25,26,27]. However, no desired edit was detected in the endogenous DMD:c.7893delC locus of EGFP+ PE2-edited fibroblasts by Sanger sequencing (Figure 6E). This observation is not completely surprising because only 1.1% prime editing efficiency was achieved in the VENUS reporter locus. It is also possible that the exogenous reporter locus is easier to be edited than the endogenous silence DMD locus because lentivirus preferred to integrate into chromatin-accessible regions.

## 4. Conclusions

Our results showed that the VENUS system provides an alternative and attractive strategy for the rapid assessment of the CRISPR-based correction of LOF mutations in cells. We observed that the CRISPR editing efficiency obtained in the exogenous VENUS reporter locus is higher than the corresponding endogenous site, which might be attributed to the epigenetic and chromatin accessibility effects. A direct comparison of the editing efficiency obtained in both reporter and endogenous loci might provide important insights into the impact of chromatin structure on CRISPR activity, and most importantly aid the further development of better CRISPR activity prediction tools. However, it should be further addressed in future studies. One limitation of the VENUS reporter system is its limited insertion size of the synthetic fragment by ligation, which could be improved by using, e.g., synthetic LOF reporter genes. However, the VENUS reporter system could probably be used as a conventional tool for assessing the effect of different modifications to the HDR template, as well as pegRNA (e.g., PBS length, RT) on CRISPR gene editing. Additionally, as demonstrated by us and several other groups, there is a good correlation of CRISPR on-target efficiency (Pearson’s R above 0.7) between the reporter and endogenous target sites [22,24,26,27]. It also provides the advantage of high throughput quantification of CRISPR gene editing outcomes when combined with lentiviral libraries.

Although not demonstrated in this study, the restoration of DMD expression can be achieved by reprogramming the CRISPR-edited fibroblasts with transdifferentiating [3]. Consistent with our previous observation [3], NHBEJ is an efficient CRISPR editing approach for restoring gene expression by predictive and in-frame deletion of nonsense mutation or frame-shift mutations. However, since NHBEJ also deletes a few extra DNA sequences in the gene, this limits broad application for gene therapy purposes. For DMD therapy, NHBEJ offers a highly efficient gene therapy strategy for partially restoring dystrophin functions. Since the NHBEJ approach uses two CRISPR gRNAs, the potential risk of an off-target effect will be increased compared to single gRNA-based HDR and PE editing strategies.

In this study, we obtained over 20% editing efficiency with HDR, which is considerably high for cells. However, this should be further evaluated in muscle tissues and is not suitable for in vivo DMD therapy, as the HDR pathway is inactive in muscle tissues. Many other factors could affect HDR efficiency in cells, such as the length of the ssODN, modifications of the nucleotides and the gRNA, form, and the time of delivering the Cas9, gRNA, and ssODN [28]. Likewise, several modifications have been reported that can increase PE efficiency in cells, such as the addition of a nicking gRNA (PE3), as demonstrated by this study, as well as the ablation of a mismatch repair [29,30], structure medication of the pegRNA to increase stability [31], and HDAC inhibition [32]. We showed that VENUS can be used to assay HDR and PE editing efficiency. Although screening gRNA by VENUS was limited by the appropriate window, the VENUS system could still be a good and conventional tool for studying, e.g., the effect of modifications on the HDR template and PE pegRNAs on CRISPR efficiency, which should be further explored in future studies.

## Figures and Tables

**Figure 1 biomolecules-13-00870-f001:**
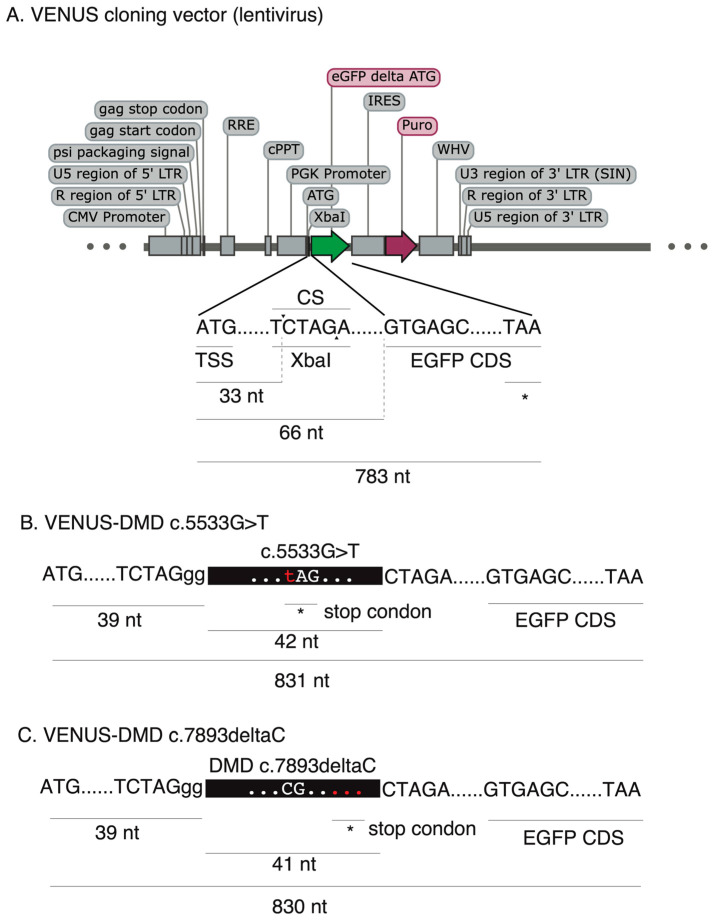
Outline of the VENUS reporter vectors. (**A**) Outline of VENUS cloning vector (lentivirus), including the translation start site (TSS), cloning site (CS, XbaI) for inserting the sequences with loss-of-function mutation, EGFP coding sequences, and the translation stop codon (asterisk, *). (**B**) The VENUS reporter vector carrying 42 base pairs of DMD coding sequencing with a stop codon mutation (c.5533G>T). (**C**) The VENUS reporter vector carrying 41 base pairs of DMD coding sequencing with one nucleotide deletion (c.7893delC).

**Figure 2 biomolecules-13-00870-f002:**
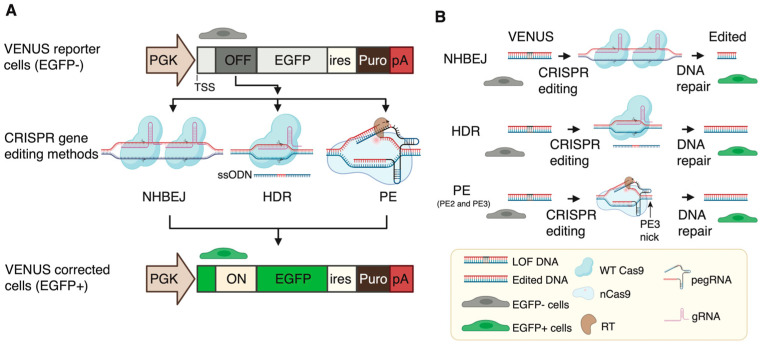
Illustration of the VENUS reporter system and three CRISPR gene editing methods. (**A**) A schematic illustration of the VENUS reporter system and the selection of three CRISPR-based systems to restore EGFP expression in cells. (**B**) A schematic illustration of the three CRISPR-based systems to restore EGFP expression through editing the inserted LOF mutation in VENUS reporter cells. TSS, translation start site; LOF, loss-of-function; OFF, inserted loss-of-function mutation, which turns off EGFP translation; PGK, phosphoglycerate kinase promoter; EGFP, enhanced green fluorescent protein gene; ires, internal ribosome entry site; Puro, gene encoding for puromycin N-acetyltransferase; pA, poly A signal; NHBEJ, non-homologous blunt end joining; HDR, homology-directed repair; PE, prime editing (PE2 and PE3 used in this study); WT, wild type; nCas9, Cas9 nickase; RT, reverse transcriptase (domain); gRNA, guide RNA; pegRNA, prime editing gRNA. The figure was partially prepared with Bioreder.com with a license for publication.

**Figure 3 biomolecules-13-00870-f003:**
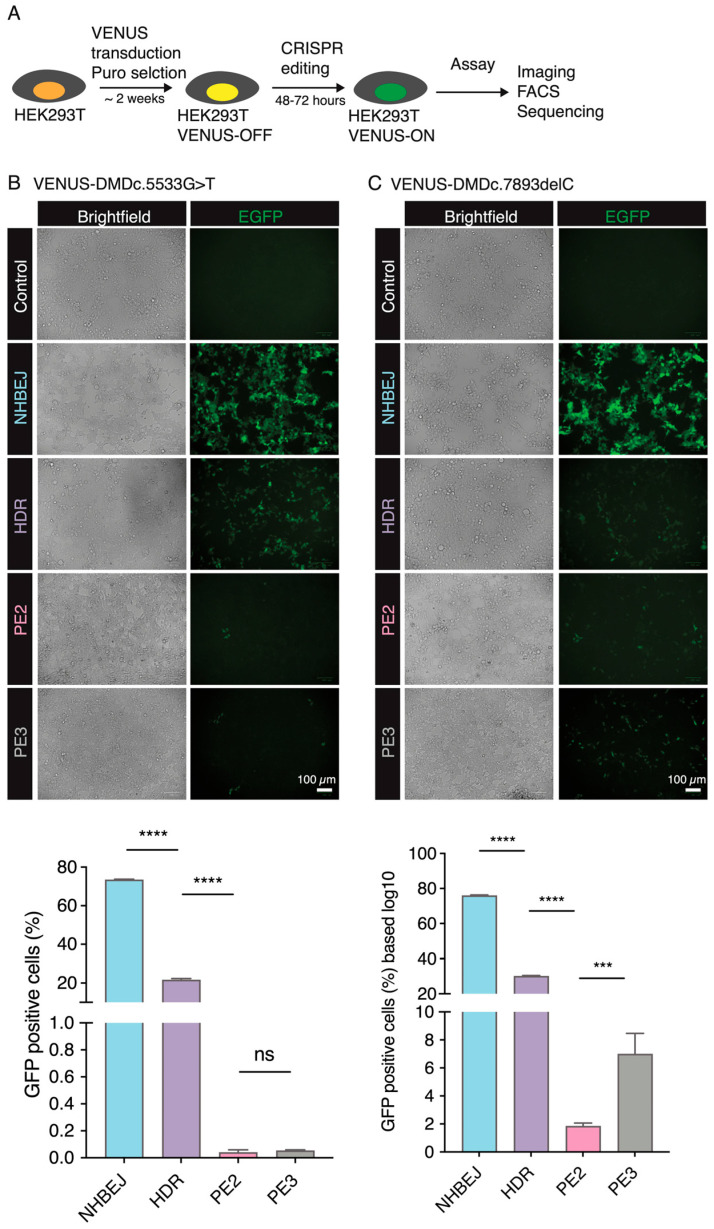
VENUS-mediated comparison of NHBEJ, HDR, and PE-mediated restoration of DMD LOF mutations in HEK293T cells. (**A**) Generation of VENUS reporter cells by the lentiviral-mediated integration of the VENUS reporter cassette into HEK293T cells. FACS, fluorescence-activated cell sorting. (**B**) Brightfield and fluorescence images in the HEK293T-based VENUS-DMD:c.5533G>T reporter cells in the unedited cells (control), and NHBEJ, HDR, PE2-, and PE3-edited cells. (Representative image from n = 3 experimental replicates.) Scale bar = 100 µm. Below: A bar plot of FACS-based quantification of EGFP+ HEK293T VENUS DMD:c.5533G>T reporter cells (n = 3 experimental replicates). Representative FACS plots are shown in Appendix A. (**C**) Brightfield and fluorescence images in the HEK293T-based VENUS DMD:c.7893delC reporter cells in the unedited cells (control), and NHBEJ-, HDR, PE2, and PE3-edited cells. (Representative image from n = 3 experimental replicates.) Scale bar = 100 µm. Below: A bar plot of FACS-based quantification of EGFP+ HEK293T VENUS DMD:c.7893delC reporter cells (n = 3 experimental replicates). Representative FACS plots are shown in Appendix A. ****, *p* < 0.0001, ***, *p* < 0.001 (one-way ANOVA).

**Figure 4 biomolecules-13-00870-f004:**
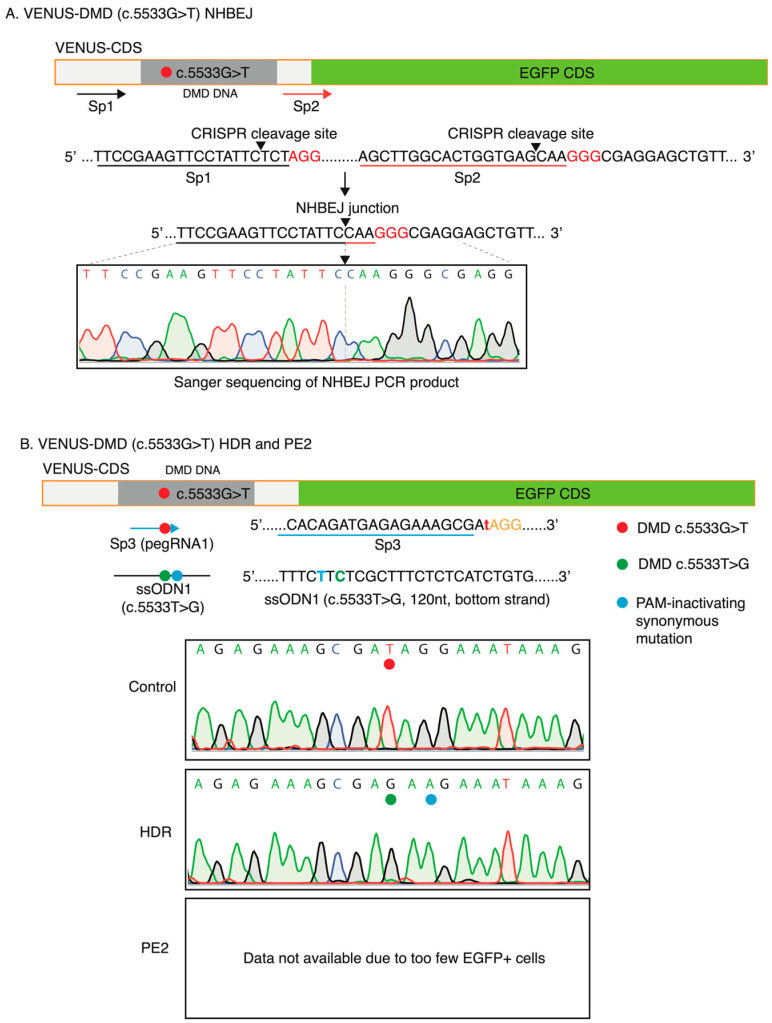
Correction of VENUS-DMD (c.5533G>T) mutation in the VENUS locus in HEK293T cells. (**A**) Validation of NHBEJ-mediated in-frame-deletion of the VENUS-DMD (c.5533G>T) mutation. Sp1 and Sp2 are gRNAs designed for NHBEJ (same as Figure 3). Expected DNA cleavage sites by the SpCas9 are indicated with an arrowhead. Theoretical NHBEJ junction is indicated. Representative Sanger sequencing result indicates the correct NHBEJ junction (n = 3 experimental replicates). (**B**) Validation of HDR-mediated correction of the VENUS-DMD (c.5533G>T) mutation in the VENUS coding sequences. Sanger sequencing for PE2 was not available due to too few EGFP+ positive cells. ssODN, single-strand oligonucleotides, which were designed with the CRISPR gRNA targeting DNA strand (120 nt). Representative (n = 3) Sanger sequencing results were shown for the control and HDR-edited cells.

**Figure 5 biomolecules-13-00870-f005:**
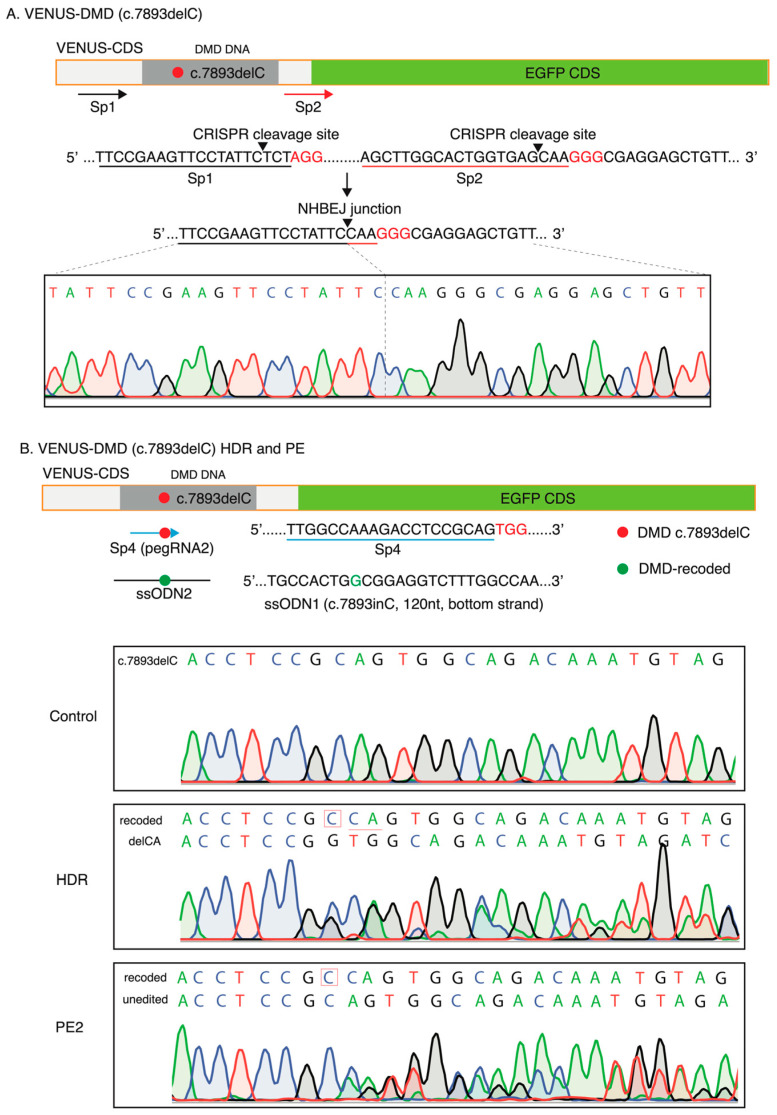
Correction of dmd:c.7893delc mutation in the venus locus in hek293t cells. (**A**) Validation of NHBEJ-mediated in-frame-deletion of the DMD:c.7893delC mutation. Sp1 and Sp2 are gRNAs designed for NHBEJ. Expected DNA cleavage sites by the SpCas9 are indicated with an arrowhead. Theoretical NHBEJ junction is indicated. Representative Sanger sequencing result indicates the correct NHBEJ junction (n = 3 experimental replicates). (**B**) Validation of HDR and PE2-mediated correction of the DMD:c.7893delC mutation in the VENUS coding sequences. ssODN, single-strand oligonucleotides, which were designed with the CRISPR gRNA targeting DNA strand (120 nt). Representative (n = 3) Sanger sequencing results were shown for the control, HDR, and PE2-edited cells. Alleles (unedited c.7893delC, recoded, and delCA) are highlighted.

**Figure 6 biomolecules-13-00870-f006:**
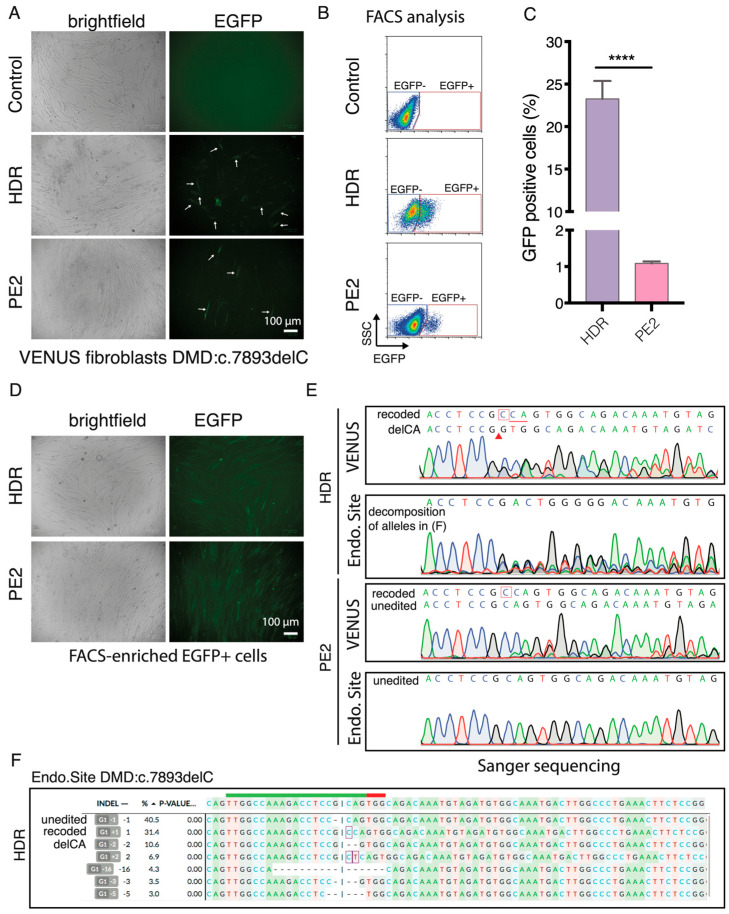
delC mutation in the VENUS locus and endogenous locus of patient fibroblasts. (**A**) Representative (n = 3 experimental replicates) brightfield and fluorescent images of the control, NHBEJ, HDR, and PE2-edited VENUS DMD:c.7893delC patient fibroblasts. Arrowheads indicate representative EGFP+ cells. (**B**) Representative FACS dot plot (n = 3) with gating for EGFP- and EGFP+ fibroblasts. SSC, side-scatter of all single cells. (**C**) Quantitation of EGFP+ cells in B (n = 3, **** represents a *p*-value < 0.0001 with a paired *t*-test). (**D**) Representative brightfield and fluorescent images of FACS-enriched cultured DMD:c.7893delC fibroblasts after HDR and PE2 editing. (**E**) Sanger sequencing chromatogram of the VENUS reporter site and the endogenous DMD:c.7893delC site (Endo.Site) in cells from D. Recoded, referring to the corrected DMD allele by inserting a C nucleotide (highlighted with a red box). delCA, referring to an indel allele with two nucleotide (CA) deletions (underline). (**F**) Decomposition of the mixed alleles in the HDR.Endo.Site in (**E**).

## Data Availability

The VENUS plasmids can be accessed at the Yonglun Luo lab at Addgene.

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
