# Peer review of "Comparison of In-Frame Deletion, Homology-Directed Repair, and Prime Editing-Based Correction of Duchenne Muscular Dystrophy Mutations"

_biomolecules, 2023, doi:10.3390/biom13050870_

Round 1
Reviewer 1 Report
In this work, the authors describe the use of different gene editing techniques to correct a mutation in a reporter construct incorporated into HEK cell DNA. The work is overall well presented, though its novelty is limited. In addition, I have a number of concerns about the design of the gene editing systems and reporters, as well as the true utility of this technique for therapeutic development:
Major:
* In the NHEJ-based correction of the mutation, the authors have targeted sequences that form part of the reporter construct rather than the mutation-containing DMD fragment inserted in that construct (which would be too small to target by this approach). Although the authors conclude that NHEJ was the most efficient editing system, the fact that the gRNA sequences designed here cannot be used at the native DMD locus precludes any meaningful further development. The same concern would apply to the position of the nicking gRNA for PE3. At the very least, the authors should point out early in the paper that these targets are located outside the DMD fragment (ideally illustrating the target locations in figure 1 or figure 2) and discuss this as a significant limitation.
* More broadly, I think the opportunities to use VENUS as a tool for screening of gRNAs and gene editing strategies is quite limited. For example, as noted above, this system cannot be used to screen gRNAs for NHEJ or nicking gRNAs for PE3. VENUS can be used to evaluate pegRNAs for PE2/PE3 and gRNAs for HDR, but this is rarely a major challenge in practice (since there are usually only a few possible gRNAs within the appropriate window to correct any given mutation). In these circumstances, work performed directly in patient-derived cells is likely both easier and more conclusive.
* The authors indicate that gene editing efficiency in reporter constructs is different when these constructs are integrated into genomic DNA (rather than episomal). However, the use of lentiviral vectors to integrate the reporter construct at essentially random locations in the genome does not guarantee that the surrounding chromatin will be in a state similar to the DMD locus. This should be noted as a limitation. Data on the overall editing efficiency in the engineered patient-derived cells at the VENUS vs native DMD loci could be helpful to assuage these concerns, but the manuscript only presents data on editing at the native locus in cells that also had editing at the VENUS locus. In this experiment, the fact that the (potentially multiple) VENUS loci show near complete editing, while editing at the native locus is ~40%, suggests that efficiency is lower at the native locus.
Minor:
* The methods section contains a few mistakes and inconsistencies in unit abbreviations (e.g. use of u instead of µ and lower case m instead of upper case M for molarity).
* The first paragraph of the results section contains some information that should be moved to the introduction (e.g. other reporter systems, background information on DMD, etc).
* Any discussion of HDR in the context of DMD (line 405) should also point out that HDR is essentially inactive in muscle tissue, and thus is not viable for in vivo therapeutic development.
* The authors suggest that VENUS is an ideal tool for developing DMD therapeutics (line 229). Most DMD-causing mutations, however, are large deletions and duplications that are not suitable for this type of reporter construct.
* The overall quality of the English language is good, but manuscript would benefit from review to correct a handful of grammar mistakes.
Author Response
(Author response starts with "Re.")
In this work, the authors describe the use of different gene editing techniques to correct a mutation in a reporter construct incorporated into HEK cell DNA. The work is overall well presented, though its novelty is limited. In addition, I have a number of concerns about the design of the gene editing systems and reporters, as well as the true utility of this technique for therapeutic development:
Re: We really appreciate the reviewer comments to our study, and the valuable suggestions for its further improvement. We agree with the reviewer that the VENUS reporter system retains its own limitations as most reporter system. But it also provides a conventional approach for rapid comparison and evaluation of gene editing efficiency and outcomes. We will definitely continue work on further improve the system and ultimately demonstrate its value in therapeutic applications.
Major:
* In the NHEJ-based correction of the mutation, the authors have targeted sequences that form part of the reporter construct rather than the mutation-containing DMD fragment inserted in that construct (which would be too small to target by this approach). Although the authors conclude that NHEJ was the most efficient editing system, the fact that the gRNA sequences designed here cannot be used at the native DMD locus precludes any meaningful further development. The same concern would apply to the position of the nicking gRNA for PE3. At the very least, the authors should point out early in the paper that these targets are located outside the DMD fragment (ideally illustrating the target locations in figure 1 or figure 2) and discuss this as a significant limitation.
Re: Thanks for the comments for the limitation of using the VENUS reporter system to capture the DMD editing efficiency in the corresponding endogenous locus. It is indeed as commented by the reviewer, as we also highlighted in our results section (line 390-391), the VENUS validated Sp1 and Sp2 gRNAs could not be used for the targeting in the endogenous DMD. We have adapted the suggestions from the reviewer and include the illustration in both Figure 4 and 5, which can better illustrate the targeting locus of the gRNAs. Limitation of the method has also been included in the discussion.
* More broadly, I think the opportunities to use VENUS as a tool for screening of gRNAs and gene editing strategies is quite limited. For example, as noted above, this system cannot be used to screen gRNAs for NHEJ or nicking gRNAs for PE3. VENUS can be used to evaluate pegRNAs for PE2/PE3 and gRNAs for HDR, but this is rarely a major challenge in practice (since there are usually only a few possible gRNAs within the appropriate window to correct any given mutation). In these circumstances, work performed directly in patient-derived cells is likely both easier and more conclusive.
Re. Thanks for this stimulating discussion. We agree that using the reporter system as a tool for screening gRNAs is limited due to the small insertion size. We have been thoroughly considering the potential limitations and advantage of such reporter systems. As now highlighted in our revised discussion, we have highlighted a few limitations of the reporter vector. Although not directly demonstrated in this study, the VENUS system could potential be used as a rapid method to assess the modifications to e.g., ssODN template, design of PE pegRNA (altering the PBS length and/or RT template). It might potentially be further developed into a pool lentiviral library based. We have taken all the valuable suggestions and included an extensive discussion in the revised manuscript.
* The authors indicate that gene editing efficiency in reporter constructs is different when these constructs are integrated into genomic DNA (rather than episomal). However, the use of lentiviral vectors to integrate the reporter construct at essentially random locations in the genome does not guarantee that the surrounding chromatin will be in a state similar to the DMD locus. This should be noted as a limitation. Data on the overall editing efficiency in the engineered patient-derived cells at the VENUS vs native DMD loci could be helpful to assuage these concerns, but the manuscript only presents data on editing at the native locus in cells that also had editing at the VENUS locus. In this experiment, the fact that the (potentially multiple) VENUS loci show near complete editing, while editing at the native locus is ~40%, suggests that efficiency is lower at the native locus.
Re. Great thanks for pointing out the different between reporter (open chromatin) vs. the endogenous locus. This is indeed a not fully addressed question in the CRISPR gene editing field. Most of the current large scale gene editing efficiency are based on genomically integrated and synthetic targeted loci. One challenge is how to evaluate the editing efficiency of the endogenous loci in large scale. Currently, our team is working addressing this challenge in other CRISPR studies. We have included this point into the discussion.
Minor:
* The methods section contains a few mistakes and inconsistencies in unit abbreviations (e.g. use of u instead of µ and lower case m instead of upper case M for molarity).
Re. All units have now been corrected.
* The first paragraph of the results section contains some information that should be moved to the introduction (e.g. other reporter systems, background information on DMD, etc).
Re. Introduction has been revised. Great thanks.
* Any discussion of HDR in the context of DMD (line 405) should also point out that HDR is essentially inactive in muscle tissue, and thus is not viable for in vivo therapeutic development.
Re. Thanks for pointing out that. We have revised it accordingly.
* The authors suggest that VENUS is an ideal tool for developing DMD therapeutics (line 229). Most DMD-causing mutations, however, are large deletions and duplications that are not suitable for this type of reporter construct.
Re. Thanks. We have now carefully revised our conclusion.
* The overall quality of the English language is good, but manuscript would benefit from review to correct a handful of grammar mistakes.
Re. We really appreciate all the insightful comments and suggestions from the reviewer. This has inspired some interesting followup development of the system.
Reviewer 2 Report
The study by Zhao et al. developed a synthetic reporter system (VENUS) to compare the efficiencies of NHBEJ, HDR and PE-based CRISPR correction of DMD mutations.
1. The title. The authors should spell out the acronym “DMD”.
2. Abstract. Line 21. It will be helpful to write one sentence or two to explain the VENUS system.
3. The last paragraph of the Introduction. The authors should give a brief introduction of DMD and the two mutations.
4. Materials and Methods. Provide catalog numbers of reagents used in this study.
5. Line 78. Spell out “ssODN”.
6. Figure 1. It will be easier for readers to understand the construction of the vectors if the authors could provide linear maps of these vectors with zoomed-in regions.
7. Figure 3B. Add scale bar(s) to the images.
8. Figure 3B. Use axis breaks and finer scales for the two plots, as they did in figure 6C.
9. Line 288-289, “… PE efficiency can be…” and the corresponding plot: is it PE2+PE3 or PE3 only. Please clarify.
10. Line 318-319. I understand the PE2 efficiency is very low (0.02%) in this case, but why couldn’t the authors get enough EGFP+ cells by sorting more cells or by expanding the sorted cells? Especially, PCR was used to amplify the target region for genotyping.
11. Figure 6A and 6D. Add scale bar(s) to the images.
12. Line 379-380. This is not unsurprising. Even though the editing in the VENUS locus and in the endogenous locus are independent events, the efficiency of editing in the endogenous locus should be a bit higher because these are EGFP+ enriched cells. The final efficiency should be a combination of efficiencies resulted from lentivirus transduction, nucleofection and genome editing.
13. From the DNA sequencing chromograms, the sanger sequencing quality in some cases seem to be suboptimal: the resolution of some single nucleotides is poor.
Author Response
Re. We thanks the reviewer’s thorough correction and evaluation of our manuscript. All suggested corrections have now been addressed in the revised manuscript.
- The title. The authors should spell out the acronym “DMD”.
Re. DMD is now spell out in the revised manuscript.
- Abstract. Line 21. It will be helpful to write one sentence or two to explain the VENUS system.
Re. Context has been provided for the VENUS in the abstract.
- The last paragraph of the Introduction. The authors should give a brief introduction of DMD and the two mutations.
Re. Introduction to the DMD and mutations have now been included in the introduction.
- Materials and Methods. Provide catalog numbers of reagents used in this study.
Re. M&M has now been revised with detail information of catalog numbers.
- Line 78. Spell out “ssODN”.
Re. ssODN has now been spell out.
- Figure 1. It will be easier for readers to understand the construction of the vectors if the authors could provide linear maps of these vectors with zoomed-in regions.
Re. Thanks for the suggestion. Figure 1 has been revised with more information.
- Figure 3B. Add scale bar(s) to the images.
Re. The scale bar is now provided.
- Figure 3B. Use axis breaks and finer scales for the two plots, as they did in figure 6C.
Re. Figure 3B bar plots have been revised to linear scale with segments.
- Line 288-289, “… PE efficiency can be…” and the corresponding plot: is it PE2+PE3 or PE3 only. Please clarify.
Re. Great thanks. This sentence has been better clarified in the revision.
- Line 318-319. I understand the PE2 efficiency is very low (0.02%) in this case, but why couldn’t the authors get enough EGFP+ cells by sorting more cells or by expanding the sorted cells? Especially, PCR was used to amplify the target region for genotyping.
Re. We did try a few times to FACS GFP+ cells for expansion. Most likely due to the reason that these are primary fibroblasts and too few cells after FACS, we did not manage to grow them up for further genotyping.
- Figure 6A and 6D. Add scale bar(s) to the images.
Re. Scale bars in the 6A and 6D were too small. We have now added new scale bar.
- Line 379-380. This is not unsurprising. Even though the editing in the VENUS locus and in the endogenous locus are independent events, the efficiency of editing in the endogenous locus should be a bit higher because these are EGFP+ enriched cells. The final efficiency should be a combination of efficiencies resulted from lentivirus transduction, nucleofection and genome editing.
Re. Thanks for the comments.
- From the DNA sequencing chromograms, the sanger sequencing quality in some cases seem to be suboptimal: the resolution of some single nucleotides is poor.
Re. The low resolution is probably due to image conversion from PDF to PNG. We will make sure that in final production article will use high resolution images.
Reviewer 3 Report
The authors compare different CRISPR-based approaches to correct loss of function mutations using an artificial system in which a small DNA fragment of DMD coding sequence containing a stop codon mutation or a1 nt deletion has been cloned between the translation start site and the eGFP ORF (called the VENUS reporter vector). They compared the efficacy of NHBEJ, HDR and prime editing to remove the stop codon. The cells used in the study are HEK293/fibroblasts stably transduced with the VENUS vector.
The study presents major weaknesses for both the HEK293 and fibroblast experiments:
-it is not realized on cellular clones but on a population of cells carrying different copies of the VENUS vector, inserted at different locations in the genome, which could introduce major bias in the study. For example, as all the experiments were not carried out in parallel, it is impossible to affirm they were realized on the same cells. Indeed, a population of transduced cells always drifts.
- the sanger sequencing is realized on genomic DNA extracted from the population of CRISPR treated cells, meaning that the sequence represents the average of the events. This is major bias in the study because this type of approach makes it difficult to highlight individual point mutations. The cloning of the PCR products and the sequencing of at least 10 different clones is required.
- How many times was realized the nucleofection? How was evaluated the nucleofection efficacy of the different approaches? For example, how do the authors eliminate the hypothesis of a low nucleofection efficiency of PE2 to explain the few eGFP+ cells? If the efficacy of the different approaches is not identical, it is not possible to compare their efficiency.
- Statistical analyses are not appropriate. Student's-T test is not recommended to compare differences between groups. The one-way analysis of variance is the appropriate method instead of the t test.
The number of eGFP + cells in figure 6 is much lower than in Figure 2. Why? How was the transfection efficacy in these 2 cell lines?
Minor
Abstract: PE2and 3 not defined. VENUS reporter cells are not explained.
Introduction: define ABE, CBE, GBE. Better explain what primer editing is and not only how it works. PE2 and PE3 must be defined in the introduction.
L69: what are the VENUS vectors? How does the system work?
L73: define LOF
Material and methods
L118 Lentivirus packaging: how many cells are seeded?
L128 how many cells are seeded?
L124 and 130: is polybrene added twice (in the medium and in the lentivirus supernatant)?
L153: what about PE3?
L264: PE2 is used in the study but PE3 is mentioned in figure 3.
L290: PE3 treated cells do not show an increase of 10 folds compared to PE2.
Author Response
The authors compare different CRISPR-based approaches to correct loss of function mutations using an artificial system in which a small DNA fragment of DMD coding sequence containing a stop codon mutation or a1 nt deletion has been cloned between the translation start site and the eGFP ORF (called the VENUS reporter vector). They compared the efficacy of NHBEJ, HDR and prime editing to remove the stop codon. The cells used in the study are HEK293/fibroblasts stably transduced with the VENUS vector.
Re. We thank all the valuable and constructive comments and suggestions from the reviewer for our study. The use of reporter vectors to capture CRISPR gene editing provides its advantages, as well as limitations, as highlighted and discussed in this manuscript.
The study presents major weaknesses for both the HEK293 and fibroblast experiments:
-it is not realized on cellular clones but on a population of cells carrying different copies of the VENUS vector, inserted at different locations in the genome, which could introduce major bias in the study. For example, as all the experiments were not carried out in parallel, it is impossible to affirm they were realized on the same cells. Indeed, a population of transduced cells always drifts.
Re. For CRISPR gene editing, unless we aim to generate single cellular clones for further applications, e.g., generation of gene edited iPSCs clones, it is better to use a population of gene edited clones to balance the clonal variations. For the VENUS cells, we did use a population of cells instead of single cell with one VENUS copy. This provides the advantage of balancing the clonal variation as well as reducing the workload in testing many clones. Indeed, we agree with the reviewer that using this mixture of clones, it is impossible to affirm all take place on the same cells. Since we are evaluating a total efficiency, we believe that this will not affect final editing readouts. One disadvantage is the uncertain number of integrated genes, which we have now discussed in the revised manuscript.
- the sanger sequencing is realized on genomic DNA extracted from the population of CRISPR treated cells, meaning that the sequence represents the average of the events. This is major bias in the study because this type of approach makes it difficult to highlight individual point mutations. The cloning of the PCR products and the sequencing of at least 10 different clones is required.
Re. For Sanger sequencing-based evaluation of CRISPR-induced indels, the outstanding algorisms developed for indel decomposition such as TIDE, ICE and DECODR have made analysis of indels from a population of CRISPR edited cells as accurate as NGS-based. Thus, with the TIDE and ICE methods, it is no longer needed to carry out the PCR product sub-cloning, colony expansion, and individual clone sequencing. We appreciate the suggestion of sequencing individually picked clones.
- How many times was realized the nucleofection? How was evaluated the nucleofection efficacy of the different approaches? For example, how do the authors eliminate the hypothesis of a low nucleofection efficiency of PE2 to explain the few eGFP+ cells? If the efficacy of the different approaches is not identical, it is not possible to compare their efficiency.
Re. In this study, we have nucleofection for NHBEJ, HDR, PE2 and PE3. All experiments have been carried in at least three experimental replicates. The beauty of using nucleofection of mRNA, RNP, as well as plus ssODN is that we can constantly achieve nearly 100%. In our previous study by Brandt et al. (https://www.mdpi.com/2218-273X/13/1/23) we have thoroughly addressed the nucleofection efficiency with mRNA in even the most difficult cells (endothelial cells) and yield nearly 100% efficiency. So the different is not due to nucleofection efficiency.
- Statistical analyses are not appropriate. Student's-T test is not recommended to compare differences between groups. The one-way analysis of variance is the appropriate method instead of the t test.
Re. Great thanks. We have included ANOVA in our multiple comparison analysis.
The number of eGFP + cells in figure 6 is much lower than in Figure 2. Why? How was the transfection efficacy in these 2 cell lines?
Re. These are not due to transfection efficiency. With our nucleofection protocol, both HEK and fibroblast can be transfected with 100%. We believe that the different in EGFP+ cells (efficiency) is more likely due to the different in DSB repair pathway preference between the cell lines.
Minor
Abstract: PE2and 3 not defined. VENUS reporter cells are not explained.
Re. Abstract has now been revised accordingly.
Introduction: define ABE, CBE, GBE. Better explain what primer editing is and not only how it works. PE2 and PE3 must be defined in the introduction.
Re. Introduction has been revised accordingly. All changes are highlighted in red font.
L69: what are the VENUS vectors? How does the system work?
Re. Extra information for the reporter vector has been provided, which is also explained in results section 1.
L73: define LOF
Re. LOF refers to loss of function mutation, which is now defined.
Material and methods
L118 Lentivirus packaging: how many cells are seeded?
Re. Number of cells is now included.
L128 how many cells are seeded?
Re. Number of cells for nucleofection is now included.
L124 and 130: is polybrene added twice (in the medium and in the lentivirus supernatant)?
Re. More information about polybrene is now included in the MM. Yes, we use this strategy to avoid the dilution of polybrene concentration when adding the lentivirus supernatant.
L153: what about PE3?
Re. PE3 is now included in the MM.
L264: PE2 is used in the study but PE3 is mentioned in figure 3.
Re. PE2 and PE3 are now updated.
L290: PE3 treated cells do not show an increase of 10 folds compared to PE2.
Re. Great thanks for pointing out this. The previous calculation is based on pair samples. We have corrected the calculation and now provided based on mean.
Round 2
Reviewer 1 Report
The authors have largely addressed my concerns and updated the manuscript accordingly. The edits however contain quite a few grammar and English language mistakes, which make them sometimes difficult to follow. I would recommend that these sections be edited to correct these mistakes and improve readability.
Author Response
We have made further English Editing.